# Parents’ Past Bonding Experience with Their Parents Interacts with Current Parenting Stress to Influence the Quality of Interaction with Their Child

**DOI:** 10.3390/bs10070114

**Published:** 2020-07-07

**Authors:** Atiqah Azhari, Ariel Wan Ting Wong, Mengyu Lim, Jan Paolo Macapinlac Balagtas, Giulio Gabrieli, Peipei Setoh, Gianluca Esposito

**Affiliations:** 1School of Social Sciences, Nanyang Technological University, Singapore 639798, Singapore; NURA0066@e.ntu.edu.sg (A.A.); WO0016NG@e.ntu.edu.sg (A.W.T.W.); mengyu.lim@ntu.edu.sg (M.L.); janp0001@e.ntu.edu.sg (J.P.M.B.); giulio001@e.ntu.edu.sg (G.G.); psetoh@ntu.edu.sg (P.S.); 2Lee Kong Chian School of Medicine, Nanyang Technological University, Singapore 639798, Singapore; 3Department of Psychology and Cognitive Science, University of Trento, 38068 Rovereto, Italy

**Keywords:** parenting stress, parental bonding, emotional availability, parent–child interaction

## Abstract

Healthy dyadic interactions serve as a foundation for child development and are typically characterised by mutual emotional availability of both the parent and child. However, several parental factors might undermine optimal parent–child interactions, including the parent’s current parenting stress levels and the parent’s past bonding experiences with his/her own parents. To date, no study has investigated the possible interaction of parenting stress and parental bonding history with their own parents on the quality of emotional availability during play interactions. In this study, 29 father–child dyads (18 boys, 11 girls; father’s age = 38.07 years, child’s age = 42.21 months) and 36 mother–child dyads (21 boys, 15 girls; mother’s age = 34.75 years, child’s age = 41.72 months) from different families were recruited to participate in a 10-min play session after reporting on their current parenting stress and past care and overprotection experience with their parents. We measured the emotional availability of mother–child and father–child play across four adult subscales (i.e., sensitivity, structuring, non-intrusiveness, non-hostility) and two child subscales (i.e., involvement and responsiveness). Regression slope analyses showed that parenting stress stemming from having a difficult child predicts adult non-hostility, and is moderated by the parents’ previously experienced maternal overprotection. When parenting stress is low, higher maternal overprotection experienced by the parent in the past would predict greater non-hostility during play. This finding suggests that parents’ present stress levels and past bonding experiences with their parents interact to influence the quality of dyadic interaction with their child.

## 1. Introduction

Healthy parent–child interactions provide opportunities for children to accrue rich social experiences for development [1,2,3,4,5,6]. These dyadic exchanges are bidirectionally influenced by nuanced patterns of emotional transactions contributed by both the parent and the child [1,2,3,4,6,7,8] and are influenced by psychological [9] and biological and factors [10,11]. The quality of parent–child interactions hinges on dyadic partners’ emotional availability to each other [12]; that is, their emotional connectedness and ability to mutually discern and respond to each other’s needs [12,13,14]. When evaluated from both the parental and child aspects of the relationship, emotionally available dyads are better able to reciprocate each others’ socio-emotional cues in a congruent manner [15,16,17].

Play is an essential activity that stimulates parent–child interactions, especially among preschool-aged children who are beginning to exercise their newfound autonomy and motor capacities [18]. Maternal and paternal interactions differ in ways that can be observed in such play situations. Through the lens of emotional availability, Bergmann and colleagues [19] showed that mothers engage in more sensitive interactions compared to fathers—an observation that was postulated to be driven by maternal predominance in caregiving functions [16,20]. Parental gender differences have also emerged in the structuring domain during play. For instance, while mothers tend to employ greater scaffolding and didactic strategies, fathers lean more towards physical play and tend to engage with their child like age-mates [14,21,22]. Although fathers customarily occupy the “playmate” role in child-rearing, father–child play has been shown to be essential for development, having been previously associated with children’s attachment security [23,24], socio-cognitive development and emotional regulation [25,26].

Parent’s attachment styles are influenced by the first relationship one has with their parents as an infant, which further influences subsequent caregiving behaviours that one may exhibit in the future. Based on attachment theory, early experiences of attachment as an infant provides exposure to social relationships that help form the internal working model of attachment, which is used as a reference for future social relationships [27]. In this way, early experiences of social relationships with a parent would shape future relationships; in particular, relationships where one becomes the caregiver. An adult’s parental attachment has been predictive of subsequent parenting behaviours [28,29]. Parental attachment, as examined by the Adult Attachment Interview, are classified into dismissive, preoccupied and autonomous [30,31]. One’s adult attachment then corresponds to subsequent parent-infant attachment, as assessed by the strange situation procedure, where mothers with autonomous adult attachment are more likely to have infants with secure attachments and mothers with preoccupied adult attachment are more likely to have infants with resistant attachments [31]. Parental attachment, in this way, would influence future parenting practices that may be captured by parental bonding history.

The prevailing literature supports the view that parental^g^ bonding history (parental^g^ denotes the intergenerational influence of past parental bonding history on parents’ current caregiving behaviours), which is evaluated from the parental care and overprotection that the parent received as a child, influence parent–child interactions in the subsequent generation [32,33,34]. Intergenerational transmission of parenting occurs when parenting practices, behaviours and attitudes of one generation are either directly or indirectly influenced by the previous generation [35]. Longitudinal studies investigating parenting behaviours across generations have revealed consistent continuities in parenting from one generation to the next [36]. Despite modest associations, intergenerational transmission of positive and negative parenting has been observed in various social and economic contexts [37,38,39,40,41,42]. Moreover, suboptimal parental bonding in a parent^g^–child relationship has been posited to be intergenerationally transmitted through less emotionally available parent–child interactions in the subsequent generation, where insecurely attached mothers exhibit less maternal sensitivity when interacting with their own children [43], while their children tend to display less responsiveness towards them [44]. Parental attachment has also been shown to be influenced by parenting stress, where mothers who perceived a lower-quality rearing from their parents were more likely to experience greater parenting stress when they become parents as compared to mothers who perceived higher-quality rearing from their parents [45].

Besides factors that reside in the past, current parenting stress inevitably affects parents’ interactions with their children too. Parenting stress reflects a state in which parenting demands exceed the coping resources that the parent possesses [46]. Higher levels of parenting stress have been consistently linked to maladaptive parenting behaviours [47,48] which are characterised by less emotionally available parenting [49,50]. For instance, mothers who experienced higher levels of parenting stress have been found to display less sensitivity [51], while exhibiting greater hostility and intrusiveness [52,53]. Parenting stress has also been associated with decreased brain-to-brain synchrony in parts of the brain that are implicated in inferring and understanding others’ mental states and social cognition, thereby adversely influencing mother–child attunement [7]. Compounding these negative effects, children of stressed and less emotionally available mothers showed less involvement and responsiveness when interacting with their mothers [52]. These studies point to the pivotal role of both past and present experiences in determining dyadic emotional availability.

Emotional availability is a construct used to assess the quality of interaction between an adult and a child. Both members of the dyad are considered as interdependent agents that contribute to their bidirectional social exchanges [12]. The Emotional Availability Scale (EAS) operationalises emotional availability into six multidimensional domains of behaviours, four of which are parental scales (i.e., sensitivity, structuring, non-intrusiveness, non-hostility) and two of which are child scales (i.e., responsiveness and involvement [12,15]. Adult sensitivity measures the extent to which the adult responds to the child in a timely and appropriate manner, which reflects his/her emotional attunement to the child. Adult structuring captures the parent’s ability to provide suitable guidance and set limits when necessary while taking the child’s autonomy into consideration. Adult non-intrusiveness assesses the parent’s proclivity to support the child’s age-appropriate autonomy by refraining from overstimulating, interfering with and controlling the interaction. The final parental scale, adult non-hostility, evaluates the absence of antagonistic responses displayed by the parent towards the child. These behaviours may take the form of disgruntled facial expressions and body language, disparaging comments and mockery and an inimical tone of voice. Covert hostility may manifest as annoyance, impatience and indifference during the interaction. Meanwhile, the child responsiveness scale characterises whether the child is genuinely eager to reciprocate the parent’s attempt at engaging with him/her, whereas the child involvement scale reflects the child’s inclination to initiate social exchanges and include the parent in the interaction. The EAS has been applied to numerous naturalistic contexts, including play situations (e.g., [16,21]) due to its property of being able to measure bilateral components of parent–child interaction. Taken together, the burgeoning literature has provided compelling evidence for the use of EAS in evaluating maternal and paternal emotional availability during play.

From the studies above, it can be seen that both parental^g^ bonding history and currently experienced parenting stress have an impact on the emotional availability of the parent when interacting with their child. Depending on the quality of parental^g^ bonding and parental stress, these changing parent-related variables will produce different levels of emotional availability during parent–child interaction. Identification of how specific patterns of parental^g^ bonding and parenting stress affect emotional availability will be clinically significant in improving the quality of the parent–child relationship [54], and provide a more nuanced approach to parenting interventions that have previously viewed parents as a relatively homogeneous group based on their life stages [55]. Psychotherapies that focus on improving parent–child interactions can take into account the prevailing relationships between parental^g^ bonding, parenting stress and emotional availability, and make use of specific strategies that target these factors to produce more efficacious outcomes [56].

However, despite the considerable literature on parental^g^ bonding history and current parenting stress (e.g., [43,44,47,48,52,53]), studies examining their simultaneous effects in both father–child and mother–child dyads is lacking. Only one study so far examined parental^g^ bonding history and current parenting stress, where parents who reported optimal parental^g^ bonding history also reported the lowest level of current parenting stress [56]. Therefore, the present study aims to investigate how parental^g^ bonding history and current parenting stress influence dyadic emotional availability observed in father–child and mother–child pairs during a typical play situation.

We embarked on this study with the following hypothesis to be tested: given the robust link between parenting stress and maladaptive parenting behaviours (e.g., [52,53,57]), we predicted that parenting stress would also interact with poor parental^g^ bonding history. However, the direction of effect of this interaction on dyadic emotional availability remains exploratory.

## 2. Methods

### 2.1. Participants

In total, 29 father–child dyads (18 boys, 11 girls; father’s age = 38.07 years, child’s age = 42.21 months) and 36 mother–child dyads (21 boys, 15 girls; mother’s age = 34.75 years, child’s age = 41.72 months) participated in this study. Participants were recruited through online platforms, such as Facebook groups and forums. The following criteria had to be met before participants were deemed eligible for the study: (1) adult participants must be at least 21 years old at the time of recruitment; (2) child participants must be between 24 and 48 months at the time of recruitment; (3) adult and child participants need to be residing in the same household in Singapore; (4) adult participants must be the biological parents of child participants; (5) all participants must not suffer from any cognitive impairments, hearing or visual impairments or major diseases that would prevent them from understanding and responding to the experimental tasks. Prior to the commencement of the study, informed consent was obtained from all participants, where parents would provide consent for their children. Participants were remunerated upon completion of the study. The study was conducted in accordance with the Declaration of Helsinki, and the protocol was approved by the Institutional Review Board of the Nanyang Technological University (IRB 2018-06-016).

### 2.2. Questionnaires

Prior to attending the laboratory sessions, adult participants were requested to complete a series of online questionnaires which consisted of basic demographic questions (e.g., birthdate), the Parental Bonding Instrument (PBI; [58]) and the Parenting Stress Index–Short Form (PSI; [59]).

*Parental Bonding Instrument (PBI)*. The PBI is a self-reported questionnaire used to assess an adult’s perception of their parent’s parenting attitudes and behaviours when they were younger [58]. It is completed retrospectively, meaning adults above the age of 16 would complete the scale in terms of how they remember their parents during their first 16 years of life. This 25-item scale measures two constructs: care (12 items) and overprotection (13 items). Examples of items on the care scale include, “Spoke to me in a warm and friendly voice,” and “Seemed emotionally cold to me”; examples of items of the overprotection scale include, “Tried to control everything I did,” and “Let me decide things for myself.” This measure has both a maternal and paternal component, to be completed for each parent respectively. Responses are measured in a 4-point scale, ranging from 0 (“very likely”) to 3 (“very unlikely”). High scores in the care sale reflect parental affection and warmth, whereas high scores in overprotection scale reflect parental control and prevention of autonomy [60,61], with test re-test coefficients scores remaining significant at *p* < 0.001 level for both care and overprotection scales over 20 years [62]. The PBI has also been found to have good validity, demonstrate high convergent validity scores with the Emotional Warmth, Rejection and Protection (EMBU) scale and have sufficient construct validity [62]. Although self-reported responses may be subjective, this construct is posited to be a valid indicator of actual parenting experiences, due to its close corroborations with siblings’ and parents’ own parenting style reports [63,64].

*Parenting Stress Index-Short Form (PSI)*. The PSI is a self-reported questionnaire assessing the amount of stress associated with parenting [59]. This 36-item measure assessed parenting stress on three 12-item subscales: parental distress, parent–child dysfunctional interaction and difficult child, and a total parenting stress index score. Parental distress assesses the extent to which parents feel competent, restricted, supported and/or depressed in their parenting roles. Parent-child dysfunctional interaction refers to the extent to which parents are satisfied with their children and their interactions with their children. Difficult child refers to the extent of difficulty in taking care of their child. Total score is an indication of the overall stress a parent is experiencing. Responses are recorded in a 5-point Likert scale, ranging from 1 (“strongly disagree”) to 5 (“strongly agree”). High scores indicate higher levels of stress. PSI has been shown to be as reliable and valid as its long form [65]. It is also known to have acceptable reliability of 0.70 for parental distress, parent-child dysfunctional interaction and difficult child, and 0.85 for total score and acceptable validity [66].

### 2.3. Procedure

The present study consisted of a home-based online questionnaire followed by a play session conducted in a standard laboratory. In the first part of the study, participants who were assessed to be eligible for the study after pre-screening would complete the online questionnaire. Each parent–child dyad was given a unique participant code which parents were required to use to identify their responses. After completion, an appointment was set for each dyad to come down to the laboratory for the second part of the study.

In the second part of the study, adult and child participants were briefed about the procedures relating to the play session. They were informed that they were being video recorded and that they may withdraw from the study at any point in time. Informed consent was obtained from the participants before they were brought to an adjacent room for the experiment. Parents were instructed to engage in a free play session with their child for a duration of 10 minutes. During the play session, the parent and child sat next to each other and a standard set of toys was provided on a table in front of them [67]. The following items were made available to the dyad: a cake and tea set, a cash register set, a doll, a toy car, two balls, a set of building blocks and three age-appropriate books. The positions of all items on the table were standardised across participants. A bell was provided for the parent in the event that they would like to terminate the play session. At the start of the session, the researchers started recording the video and left the room, only to return at the end of the 10-minute mark. Participants were then debriefed and remuneration was provided. The dataset generated for this publication are available on the Data Repository of the Nanyang Technological University at the following page https://doi.org/10.21979/N9/IZQPBI [68].

### 2.4. Play Coding

*Emotional Availability Scale (EAS)*. The EAS measure assesses the emotional quality of a parent–child relationship [69]. This is done by rating the emotional availability of parent–child interactions through four global adult scales (i.e., adult sensitivity, adult structuring, adult non-intrusiveness, adult non-hostility) and two global child scales (i.e., child responsiveness and child involvement). Two independent coders were trained on EAS scoring prior to coding the recorded videos. They rated each of the six dimensions on a Likert scale ranging from 1 to 7. The “irr” package in RStudio [70] was used to calculate inter-rater agreement between the two coders. An inter-rater agreement of at least 80% was achieved for each EAS subscale.

In the event where scores between the two coders were different, the two coders deconflicted their ratings and agreed upon a final score. The EAS has also been shown to be highly reliable, with satisfactory construct validity [15].

### 2.5. Analytical Plan

#### 2.5.1. Descriptive Analyses

The mean and standard deviations of all EAS, PBI and PSI scores, along with the age of the participants, were reported.

#### 2.5.2. Preliminary Analyses

Preliminary analysis of variance (ANOVA) analyses were conducted to examine the effects of the child’s sex and age and the parent’s age on EAS, PSI and PBI scores in pooled father–child and mother–child samples.

Inter-correlation coefficients between different subscales of EAS, PSI and PBI were reported.

#### 2.5.3. Inferential Analyses

To test the separate and interacting effects of parenting stress and parental^g^ bonding history on emotional availability in a parent–child interaction, analysis of variance (ANOVA) with EAS scores as the dependent variable, and PSI and PBI as the independent variable, were conducted on the pooled sample consisting both mother–child and father–child dyads. Parental gender was included as a factor in all models (e.g., non-hostility ~ parenting distress * maternal^g^ overprotection * parental gnder). Since all combinations of PBI and PSI scales were being tested, this resulted in 16 models for each EAS scale. As such, Bonferroni correction was applied, such that the new alpha was 0.05/16 = 0.00313. All statistical analyses were conducted on R studio (version 1.0.153, R-core 3.4.2).

## 3. Results

### 3.1. Descriptive Results

Descriptive data for the pooled sample, along with data on father–child and mother–child samples, are reported in Table 1.

### 3.2. Preliminary Results

Preliminary analyses of the pooled father–child and mother–child samples did not show a significant main effect of child’s sex, child’s age or parent’s age on any of the dependent and independent variables.

Inter-correlation coefficients between the different subscales of EAS (i.e., adult sensitivity, adult structuring, adult non-intrusiveness, adult non-hostility, child responsiveness, child involvement), the subscales of PBI (i.e., matercal care, maternal overprotection, paternal care, paternal overprotection) and PSI (parental distress, difficult child, parent–child dysfunctional interaction and total parenting stress) are reported in Table 2.

### 3.3. Inferential Results

Inferential analyses of the pooled father–child and mother–child samples showed a significant main effect of parental gender on adult non-intrusiveness across all non-intrusiveness models: parental distress by maternal^g^ care model (F(1,54) = 18.156; *p* = 8.2 × 10^−5^), parental distress by maternal^g^ overprotection model (F(1,54) = 16.004; *p* = 0.000194), parental distress by paternal^g^ care model (F(1,54) = 14.938; *p* = 0.0003), parental distress by paternal^g^ overprotection model (F(1,54) = 14.019; *p* = 0.000441), parent–child dysfunctional interaction by maternal^g^ care model (F(1,54) = 19.504; *p* = 4.86 × 10^−5^), parent–child dysfunctional interaction by maternal^g^ overprotection model (F(1,54) = 17.365; *p* = 0.000112), parent–child dysfunctional interaction by paternal^g^ care model (F(1,54) = 15.595; *p* = 0.000229), parent–child dysfunctional interaction by paternal^g^ overprotection model (F(1,54) = 13.753; *p* = 0.000493), difficult child by maternal^g^ care model (F(1,54) = 19.865; *p* = 4.23e-05), difficult child by maternal^g^ overprotection model (F(1,54) = 17.010; *p* = 0.000129), difficult child by paternal^g^ care model (F(1,54) = 16.145; *p* = 0.000183), difficult child by paternal^g^ overprotection model (F(1,54) = 14.847; *p* = 0.000312), total parenting stress by maternal^g^ care model (F(1,54) = 18.912; *p* = 6.11 × 10^−5^), total parenting stress by maternal^g^ overprotection model (F(1,54) = 17.319; *p* = 0.000114), total parenting stress by paternal^g^ care model (F(1,54) = 15.615; *p* = 0.000227), total parenting stress by paternal^g^ overprotection model (F(1,54) = 13.800; *p* = 0.000484). Post-hoc *t*-test analyses demonstrated that fathers showed significantly greater non-intrusiveness compared to mothers (t(59.94) = 4.26; *p* < 0.001; Figure 1).

ANOVA analyses also elicited a significant interaction between difficult child score and maternal^g^ overprotection on adult non-hostility (F(1,54) = 9.963; *p* = 0.00261). Regression slope tests of maternal^g^ overprotection, averaged across difficult child scores, showed a significant difference between mean − 1 SD, mean and mean + 1 SD values of maternal^g^ overprotection (t ratio = 3.494; *p* = 0.0009; Figure 2). From Figure 2, differences in maternal^g^ overprotection could be observed at lower values of difficult child score. A contrast between mean − 1 SD and mean + 1 SD maternal^g^ overprotection score yielded a significant difference when difficult child score = 0, further verifying this observation (t ratio = 3.824; df = 58; estimate = 2.75; *p* = 0.0003). When difficult child scores are low, higher maternal^g^ overprotection would result in greater adult non-hostility compared to lower maternal^g^ overprotection.

## 4. Discussion

This study seeks to investigate the link between parenting stress and parental^g^ bonding history (i.e., parental care and overprotection that a parent received as a child) on the emotional availability in a parent–child interaction. We embarked on this study with one hypothesis: that parenting stress would interact with parental^g^ bonding history to influence emotional availability. The hypothesis was supported. In the pooled mother–child and father–child sample, greater adult non-hostility, a component of emotional availability, was observed when parents report low parenting stress on the difficult child subscale, but only for parents who perceived higher maternal^g^ overprotection. These results suggest that greater perceived parental^g^ overprotection, when combined with lower parenting stress, predict enhanced emotional availability.

Findings from the pooled mother–child and father–child samples revealed that parents displayed less hostile parenting behaviours when they believe their child to be manageable, but only if they experienced greater maternal overprotection in their childhood. This phenomenon is largely contradictory to the literature on the perception of maternal overprotection, which has been shown to have suboptimal outcomes for the child [71]. This may be attributed to cultural differences, where maternal overprotection may be more positively received in Asian families. In a study on immigrant Chinese mothers’ parenting styles, it was found that authoritative parenting was associated with fewer adjustment problems in children [72]. Stright and Yeo [73], in studying Singaporean children’s perception of their mother’s parenting styles, found that their perception of maternal use of psychological control was positively associated with perception of maternal warmth. Moreover, perception of greater parental control as a function of order-keeping (i.e., parental organisation) was found to be positively correlated with greater independence and self-esteem in children [74]. Maternaloverprotection, in this way, may have captured maternal^g^ control in childhood that was serving as a way to enforce well-defined limits for children’s behaviours, allowing for children to practise autonomy through controlling their own behaviours. Additionally, maternal control in Asian mothers is typically expressed in situations involving difficult child behaviours, which is in accordance with the use of parental behavioural control to encourage impulse control and proper conduct in children [75,76]. Perception of maternal control may then be associated with greater capabilities in stress management by fostering a sense of competence and autonomy in their child. It is possible that through this, parents who perceived greater parental^g^ overprotection when younger may have developed stress coping mechanisms that were beneficial in their parenting roles, which would explain the lower difficult child and parenting stress scores, as observed in this study. Travis and Combs-Orme [77] also found similar results, in that mothers who received poor parenting in childhood (i.e., reported high maternal^g^ overprotection) were able to report lower levels of parenting stress and developed greater resilience to life stressors. They also reported low levels of dissatisfaction with their parent–child interaction and low levels of difficulty in managing their child, much like positive-adaptive mothers (mothers who recollected warm and non-controlling parenting in childhood) in the study. In this way, mothers who perceived greater maternal^g^ overprotection may have found their child manageable, and thus display fewer hostile behaviours.

Conversely, results also indicate that lower adult non-hostility scores were observed when greater maternal overprotection scores interacted with greater difficult child scores. This suggests that the advantage of parental non-hostility diminishes when parents perceive both greater maternal overprotection and greater stress in child management (as denoted by higher difficult child scores). High difficult child scores indicate a lower perceived ability to cope with parenting stress that stems from strenuous child management [59]. Although maternal overprotection may have been protective in parents with low difficult child scores (i.e., greater non-hostility observed in parenting behaviours), it is possible that maternal overprotection loses that effectiveness in parents who find their children difficult to manage in the first place. For example, children with difficult temperaments have been shown to covary with greater maternal intrusiveness [78], lower maternal sensitivity [79] and greater parenting stress [80]. When parental coping resources are overwhelmed in this way, parents then rely on prior experiences (i.e., parenting practices received when younger). Indeed, studies on intergenerational transmission of parenting showed that parents who perceived greater maternal^g^ control when younger are more likely to repeat similar power-assertive discipline methods when they become parents, especially so if they perceived their children as difficult to manage [37,81]. Through this, parenting stress associated with child management is positively associated with maternal^g^ overprotection parents received when they were younger, such that parents with higher difficult child stress display lower non-hostility scores as compared to parents with lower difficult child stress.

Additionally, it should be noted that non-hostility as measured in the emotional availability scale is scored based on the absence of hostile responses, whether they are covert or openly hostile. It is an indicator of background anger where the adult’s hostility does not need to be directed at the child [15], in which case, displays of dissatisfaction, impatience, anger, etc., are also included in the scoring of hostile behaviours during a parent–child interaction. Non-hostility scores have also been found to be strongly correlated with maternal stress [52]. Indeed, mothers who are more attuned to their children’s emotional cues and expressions are better able to respond to their children’s needs and respond appropriately, whereas mothers who perceive their children to be harder to understand have a higher tendency to be less emotionally present in their interactions with their child, which may be displayed as speaking in flat tones and less ideal response timing [53,82]. Put together, these studies illustrate how maternal^g^ overprotection may influence or interact with current experience of parenting stress (particularly in child management), which together have the combined influence on a parent’s degree of hostility in parenting behaviours.

As with all studies, interpretation of the results obtained has to be considered in the context of its limitations. Access to actual parenting behaviours that parents in this study received in childhood are not available. This means that responses on parental^g^ bonding history are retrospective and vulnerable to interpretive biases. Recollections of parenting received in childhood may be influenced by a participant’s present status at the time of responding (e.g., [83]). This may have had an influence on the way participants recalled parenting in childhood, which might explain the contrary results obtained with regard to maternal^g^ and paternal^g^ overprotection, current participants’ emotional availability and parenting stress. In addition, the exact mechanisms behind the results obtained are also uncertain because the results of this study are correlated in nature. It is possible that the parental^g^ bonding history may have a different relationship with current parenting behaviours and parenting stress. For example, perception of poor paternal^g^ parenting has been hypothesised to lead to better paternal parenting in subsequent generations based on the reworking hypothesis [84]. Men who reported receiving less warm parenting from their fathers in childhood are more likely to report experiencing the most parenting stress and rated themselves worse on paternal ability. This, in turn, was associated with greater time spent in engaging their children in verbal stimulation and physical play. This is an avenue that merits further investigation, for example, by conducting a longitudinal study in examining actual parenting behaviours as received by parents when they were children and its effects on current parenting behaviours. Future studies may also explore the use of the PBI to investigate cultural differences in the use and understanding of parental^g^ overprotection and control. As seen in this paper, parental^g^ overprotection may not necessarily have negative implications in the local context.

The finding that parental^g^ overprotection, rather than parental^g^ care, has a relationship with the frequency of hostile and sensitive parenting behaviours, requires further investigation. It is likely that a highly warm and highly overprotective parental^g^ bonding history may allow for a more positive socio-emotional development in children. Parenting in childhood that is high in warmth and control has also been shown to be associated with children’s greater ability to regulate their behaviour and attention [72]. Alternatively, it is also possible that a highly overprotective parental^g^ bonding history served as a reminder for current parents to do better, allowing for the development of more emotionally available parenting.

Additionally, the implications of this study fill a gap in the literature in understanding how parental^g^ bonding history may interact with current parenting stress to influence emotional availability in dyadic interactions. It highlights possible areas of misconception that need to be addressed with regard to the effects of the father’s involvement in parenting and provide evidence on the differential effects of gender on the parent–child relationship. These results provide a backdrop for future studies to examine the associations between current parenting behaviours and experiences and past parental^g^ experiences. Understanding these directional relationships, and the parenting attitudes that may have developed from parental^g^ bonding history, can be the next steps to take in elucidating parenting behaviours in Asian societies.

Finally, taking the results and implications discussed above, the findings of this study may be able to shed light on clinical applications on parenting skills and parent–child relationships in a culturally-specific context. It must be noted that parents from cultures where parental overprotection (or specifically in this study, maternal overprotection) is perceived more positively [73] may not benefit from parenting strategies that focus on cultivating child autonomy and independence at the expense of decreasing parental protection, as it disrupts the larger cultural influences of encouraging greater parental protection and involvement [85]. Instead, a more nuanced approach in teaching parenting skills that have to do with setting developmentally appropriate limits without undermining the child’s potential to develop autonomy and independence [86] would be recommended.

## 5. Conclusions

To conclude, current parenting behaviours may be influenced by both past and present circumstances. This means that when examining the development of parenting behaviours, both parenting received as a child (i.e., parental^g^ bonding history) and current parenting stress experienced should be taken into account. When parenting behaviours are assessed in terms of dyadic emotional availability, the bidirectional relationship between the parent and child are taken into account. Use of parental^g^ bonding history and parenting stress to understand parenting behaviours as assessed by the emotional availability scale in this study hints not only at the development of parenting behaviours, but also at how past parenting received and current parenting stress influence current parent–child interaction. When seen in this way, parental^g^ bonding history and parenting stress have long lasting effects on parent–child interactions and relationships. Alternatively, it may be that the parent–child relationship may have had an influence on the way parenting stress and current parenting behaviours are demonstrated and experienced. Reciprocal interactions from child to parent have a significant influence on how parenting stress is experienced, as evident in the influence of the difficult child subscale on the interaction between parental^g^ bonding history and current parenting behaviours. All in all, this study showed that in studying current parenting behaviours, several factors may need to be taken into account: past parenting received, current parenting stress and relationships with one’s parent and one’s child.

## Figures and Tables

**Figure 1 behavsci-10-00114-f001:**
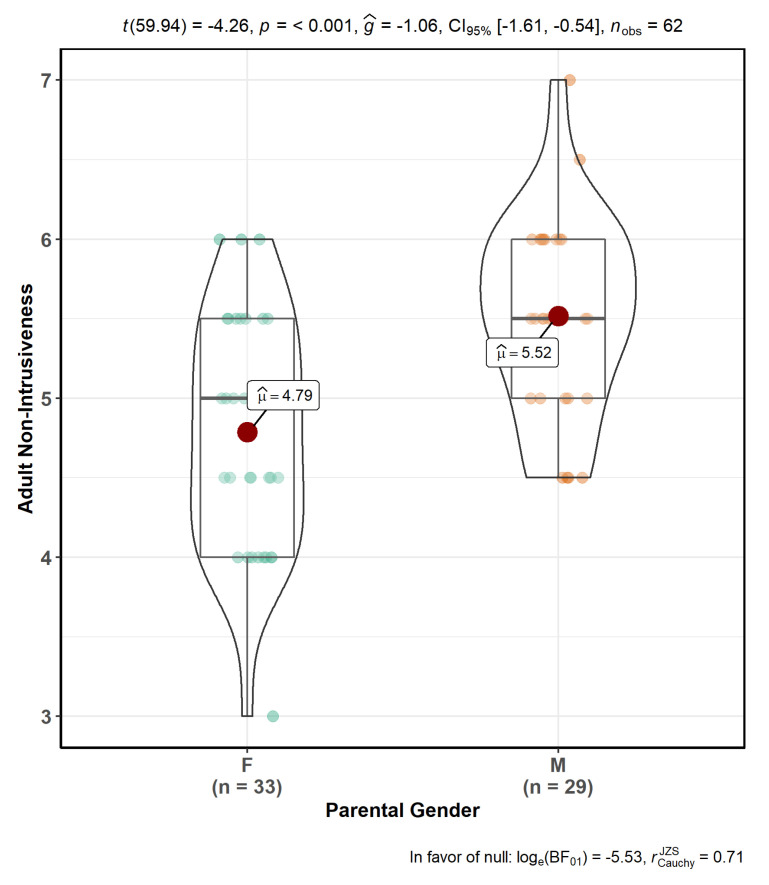
Violin plot which depicts fathers’ greater non-intrusiveness compared to mothers’ during play sessions.

**Figure 2 behavsci-10-00114-f002:**
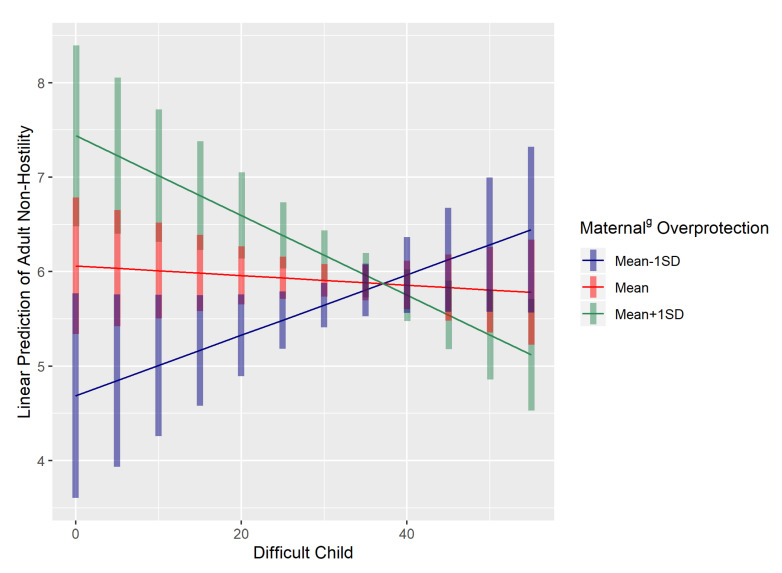
Difficult child scores predict adult non-hostility, moderated by maternal^g^ overprotection levels. At lower values of difficult child score, higher maternal^g^ overprotection predicted greater adult non-hostility.

**Table 1 behavsci-10-00114-t001:** Descriptive data for pooled father–child and mother–child samples (Parents), for the father–child sample (Father), and for the mother–child sample (Mother). Notes. *M* (*SD*) = mean (standard deviations).

	Parents	Father	Mother
	*M* (*SD*)	*M* (*SD*)	*M* (*SD*)
Age	36.34 (4.27)	38.07 (3.67)	34.82 (4.22)
Child’s Age	42.10 (5.67)	42.21 (5.25)	42.00 (6.09)
Parental Bonding Instrument (PBI)
Maternal^g^ Care	23.40 (7.53)	25.21 (6.88)	21.82 (7.83)
Maternal^g^ Overprotection	12.90 (5.74)	13.10 (4.65)	12.73 (6.62)
Paternal^g^ Care	17.16 (8.84)	15.07 (8.35)	19.00 (8.97)
Paternal^g^ Overprotection	11.73 (6.83)	9.31 (5.53)	13.85 (7.22)
Parental Stress Index	87.05 (8.85)	87.93 (18.52)	86.27 (19.39)
Parenting Distress	31.87 (9.24)	32.76 (9.92)	31.09 (8.68)
Parent-Child Dysfunctional Interaction	23.34 (6.42)	23.41 (5.77)	23.27 (7.03)
Difficult Child Score	31.84 (7.91)	31.76 (8.13)	31.91 (7.84)
Emotional Availability Scales (EAS)
Adult Sensitivity	5.32 (0.77)	5.38 (0.83)	5.27 (0.73)
Adult Structuring	5.21 (0.70)	5.24 (0.65)	5.18 (0.76)
Adult Non-Intrusiveness	5.13 (0.77)	5.52 (0.62)	4.79 ( 0.73)
Adult Non-Hostility	5.82 (0.71)	5.95 (0.66)	5.71 (0.75)
Child Responsiveness	5.43 (0.81)	5.57 (0.78)	5.30 (0.84)
Child Involvement	5.35 (0.98)	5.62 (0.70)	5.12 (1.13)

**Table 2 behavsci-10-00114-t002:** Inter-correlation coefficients between different subscales of the Parenting Stress Index (PSI), the Parental Bonding Instrument (PBI) and the Emotional Availability Scale (EAS). Notes. *PD* = parental distress, *PCDI* = parent-child dysfunctional interaction, *DC* = difficult child, *Total PSI* = total parenting stress score, *Care_M_* = maternal care, *Overprotect_M_* = maternal overprotection, *Care_P_* = paternal care, *Overprotect_P_* = paternal overprotection, *Sensitivity* = adult sensitivity, *Structuring* = adult structuring, *Non-Intrusive* = adult non-intrusiveness, *Non-Hostile* = adult non-hostility, *Responsive* = child responsiveness, *Involved* = child involvement. * *p* ≤ 0.05, *** *p*≤ 0.001 (Bonferroni corrected).

	PD	PCDI	DC	Total PSI	Care_M_	Overprotect_M_	Care_P_	Overprotect_P_	Sensitive	Structuring	Non-Intrusive	Non-Hostile	Responsive	Involved
PD	-	-	-	-	-	-	-	-	-	-	-	-	-	-
PCDI	0.48 *	-	-	-	-	-	-	-	-	-	-	-	-	-
DC	0.32	0.61 ***	-	-	-	-	-	-	-	-	-	-	-	-
Total PSI	0.79 ***	0.83 ***	0.78 ***	-	-	-	-	-	-	-	-	-	-	-
Care_M_	−0.27	−0.24	−0.05	−0.24	-	-	-	-	-	-	-	-	-	-
Overprotect_M_	0.26	0.48	0.27	0.40	−0.10	-	-	-	-	-	-	-	-	-
Care_P_	−0.18	−0.04	0.07	−0.08	0.38	0.01	-	-	-	-	-	-	-	-
Overprotect_P_	0.12	0.43	0.21	0.30	−0.19	0.80 ***	0.16	-	-	-	-	-	-	-
Sensitive	−0.01	−0.13	−0.09	−0.09	−0.09	0.11	−0.09	−0.01	-	-	-	-	-	-
Structuring	−0.05	−0.21	−0.13	−0.15	0.02	0.12	−0.01	0.04	0.72 ***	-	-	-	-	-
Non-Intrusive	0.21	0.16	0.12	0.21	−0.09	0.11	−0.14	−0.12	0.33	0.18	-	-	-	-
Non-Hostile	−0.03	−0.26	−0.11	−0.15	0.06	0.17	−0.14	0.00	0.76 ***	0.61 ***	0.20	-	-	-
Responsive	0.15	0.04	−0.01	0.08	−0.10	0.26	−0.09	0.05	0.58 *	0.47	0.53	0.36	-	-
Involved	0.08	−0.03	0.00	0.03	−0.03	0.17	0.04	−0.05	0.39	0.32	0.46	0.22	0.76 ***	-

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
