# Peer review of "Parents’ Past Bonding Experience with Their Parents Interacts with Current Parenting Stress to Influence the Quality of Interaction with Their Child"

_behavsci, 2020, doi:10.3390/bs10070114_

Round 1

Reviewer 1 Report

            “Parents’ past bonding experience with their parents interacts with current parenting stress to influence quality of interaction with their child” tests whether retrospective reports of their own parents’ behavior predict current play interactions with their children, moderated by current parenting stress (at times partitioned by parent gender).

            This paper has many strengths. This study’s greatest asset is its behavioral measurement of parent-child interaction. This high-impact, rigorous method combined with high-quality behavioral coding is very important for understanding parenting dynamics. Another strength of the manuscript is its figures, which make the interaction analyses very clear to the reader. I believe these strengths would be clearer if the methods and analyses section included greater detail.

            Primarily, I would like more information about the inter-correlations between different factors of the PBI, PSI & EAS. A correlation table providing the association between all subscales would improve the manuscript. Given that these subscales are being compared against each other, it is important for the reader to understand the association between subscales. If the subscales of each measure are strongly associated with each other, it may also be more parsimonious to aggregate each of these scales. Given the number of potential predictors, outcomes and moderators, it will also give the reader confidence that the authors are not finding false positives due to inflated type one error rate. Additionally, the authors should include the inter-rater reliability of the observed play variables.  

            Finally, I do not believe the authors have the necessary statistical power to partition these analyses by gender. The null results of the 29 father-child dyads could certainly be the result of type-2 error. Additionally, a true comparison of mothers and fathers would require a three-way interaction of gender, bonding history and parental stress predicting parent-child interactions (although power is also a concern here). I believe that paper, especially the discussion, would be stronger without the gender comparisons, which I do not have overwhelming confidence in.

            Finally, as a minor point, I believe the manuscript would be strengthened by a more overt discussion of attachment theory. Parental bonding history has significant overlap with adult attachment (as measured by the Adult Attachment Interview). There is quite a bit of literature on the interaction between adult attachment and stress predicting parenting. I would recommend especially two meta-analyses by Verhage et al. (2016 & 2018).

Verhage, M. L., Fearon, R. P., Schuengel, C., van IJzendoorn, M. H., Bakermans‐Kranenburg, M. J., Madigan, S., ... & Mangelsdorf, S. (2018). Examining ecological constraints on the intergenerational transmission of attachment via individual participant data meta‐analysis. Child Development, 89, 2023-2037.

Verhage, M. L., Schuengel, C., Madigan, S., Fearon, R. M., Oosterman, M., Cassibba, R., ... & van IJzendoorn, M. H. (2016). Narrowing the transmission gap: A synthesis of three decades of research on intergenerational transmission of attachment. Psychological Bulletin, 142, 337.

Author Response

Dear Reviewers,

Thank you for kindly reviewing our manuscript and for providing your invaluable feedback. Please find our response to each of the comments below.

Best regards,

Gianluca

REVIEWER 1

 “Parents’ past bonding experience with their parents interacts with current parenting stress to influence quality of interaction with their child” tests whether retrospective reports of their own parents’ behavior predict current play interactions with their children, moderated by current parenting stress (at times partitioned by parent gender).

           This paper has many strengths. This study’s greatest asset is its behavioral measurement of parent-child interaction. This high-impact, rigorous method combined with high-quality behavioral coding is very important for understanding parenting dynamics. Another strength of the manuscript is its figures, which make the interaction analyses very clear to the reader. I believe these strengths would be clearer if the methods and analyses section included greater detail.

           Primarily, I would like more information about the inter-correlations between different factors of the PBI, PSI & EAS. A correlation table providing the association between all subscales would improve the manuscript. Given that these subscales are being compared against each other, it is important for the reader to understand the association between subscales. If the subscales of each measure are strongly associated with each other, it may also be more parsimonious to aggregate each of these scales. Given the number of potential predictors, outcomes and moderators, it will also give the reader confidence that the authors are not finding false positives due to inflated type one error rate. Additionally, the authors should include the inter-rater reliability of the observed play variables. 

>> Thank you for your kind feedback and for these excellent suggestions. Following your recommendation, we have run inter-correlations between the different subscales, which we have incorporated in the revised manuscript as Table 2. However, we note that only the PSI subscales are strongly correlated with each other, whereas the subscales of the PBI and EAS subscales do not consistently show strong associations. As such, we thought that it would be better to keep the analyses of the PBI and EAS subscales separate.

As for PSI, the scale consists of 3 subscales (PD, DC, PCDI) and a total parenting stress index score (sum of the three subscales). We have further clarified in the manuscript that there are essentially four components of PSI, including the total parenting stress index score. Since, the total parenting stress index has been analysed in the previous set of analyses, we did not perform an aggregation of the PSI subscales.

We have also reported inter-rater agreement between coders of at least 80% for each EAS subscale. Inter-rater agreement was computed using the “irr” package on RStudio statistical computing software.

           Finally, I do not believe the authors have the necessary statistical power to partition these analyses by gender. The null results of the 29 father-child dyads could certainly be the result of type-2 error. Additionally, a true comparison of mothers and fathers would require a three-way interaction of gender, bonding history and parental stress predicting parent-child interactions (although power is also a concern here). I believe that paper, especially the discussion, would be stronger without the gender comparisons, which I do not have overwhelming confidence in.

>> We agree that the null results obtained in the father-child sample could be due to type-2 error from the partitioning of analyses by gender. Following your recommendation, we have removed the separate analyses on mother-child and father-child samples, and have only retained the analyses of the pooled sample. We also concur that a true comparison of mothers and fathers would require a three-way interaction by parental gender, which we have conducted previously.

           Finally, as a minor point, I believe the manuscript would be strengthened by a more overt discussion of attachment theory. Parental bonding history has significant overlap with adult attachment (as measured by the Adult Attachment Interview). There is quite a bit of literature on the interaction between adult attachment and stress predicting parenting. I would recommend especially two meta-analyses by Verhage et al. (2016 & 2018).

>> We agree that attachment theory would provide greater depth and has added a paragraph introducing its relation with parental bonding history as suggested.

“Parental attachment styles are influenced by the first relationship one has with their parents as an infant, which further influences subsequent caregiving behaviours that one may exhibit in the future. Based on the attachment theory, early experiences of attachment as an infant provides exposure to social relationships that helps form the internal working model of attachment, which is used as a reference for future social relationships [27]. In this way, early experiences of social relationships with a parent would shape future relationships, in particular relationships where one becomes the caregiver. Adult’s parental attachment has been predictive of subsequent parenting behaviours [28,29]. Parental attachment, as examined by the Adult Attachment Interview, are classified into Dismissing, Preoccupied and Autonomous [30,31]. One’s adult attachment then corresponds to subsequent parent-infant attachment as assessed by the Strange Situation Procedure, where mothers with Autonomous adult attachment are more likely to have infants with Secure attachments and mothers with Preoccupied adult attachment are more likely to have infants with Resistant attachments [31]. Parental attachment, in this way, would influence future parenting practices that may be captured by parental bonding history.”

Reviewer 2 Report

The authors aim to examine the influence of the interaction between parents’ past bonding experience with their parents and current parenting stress on the quality of interaction with their child. Healthy dyadic interactions involve mutual emotional availability of both the parent and child and are essential for child development. Whether parents’ emotional availability is influenced by their bonding history with their own parents and their current parenting stress has not been previously explored. The present study involved 29 father-child dyads and 36 mother-child dyads and evaluated their emotional availability during 10-min play session as well as current parenting stress and parents’ past care and overprotection experience. The manuscript addresses two important questions and uses appropriate statistical analyses to test the hypotheses. Findings from the pooled father-child and mother-child samples across all models incorporating bonding history and parenting stress indicated that fathers exhibited greater non-intrusiveness compared to mothers. In addition, regression slope analyses of the pooled sample showed that when difficult child score is low, higher maternal overprotection would predict greater adult non-hostility. In the mother-child sample, slope analyses showed that when difficult child score or parenting stress is low, higher paternal overprotection would predict greater adult sensitivity. The authors dealt with the discrepancy between these results and those of the existing literature by attributing it to cultural differences and also suggested alternative mechanisms for these interactions. Overall, the manuscript is written well and would benefit the field of parenting research.

Abstract:

  1. Lines 15-16: “Regression slope analyses showed that when parenting stress is low, higher parental overprotection would predict greater adult non-hostility”. Please be more accurate here, if this refers to the pooled sample “parenting stress” should be replaced with “difficult child score”, and “parental overprotection” with “maternal overprotection” as describe in the results.

Introduction:

  1. Line 90-91: “Despite such extensive literature….”. Please rewrite the sentence to clarify what “such extensive literature” refers to.
  2. I would add a paragraph highlighting the importance of identifying bonding history and parenting stress as predictors for emotional availability in the sense that when a parent experiences difficulties (i.e., poor bonding history and/or high parenting stress) these can be targeted by psychological therapies and thereby can improve emotional availability when interacting with the child. 

Discussion:

  1. Another paragraph should be added to suggest an explanation why overprotection was beneficial only in low score of difficult child and parenting stress. In other words, why is it different in higher scores where overprotection does not play a role in the association between stress and adult non-hostility/sensitivity.
  2. I would add a few sentences about the potential of psychological therapies in the cultural context in helping parents to resolve their issues around parental overprotection and thereby improve non-hostility/sensitivity when playing with their child. Likewise, parents who experience high parenting stress or perceive their child as difficult, regardless their bonding history, may also benefit from therapies aiming to reduce their stress by increasing their emotional availability.

The manuscript would benefit from a close edit to correct typos and formatting and language errors (including symbols, numbers, punctuation, etc). For example, throughout the text many words are followed by the letter “g”.

Author Response

Dear Reviewers,

Thank you for kindly reviewing our manuscript and for providing your invaluable feedback. Please find our response to each of the comments below.

Best regards,

Gianluca

REVIEWER 2

The authors aim to examine the influence of the interaction between parents’ past bonding experience with their parents and current parenting stress on the quality of interaction with their child. Healthy dyadic interactions involve mutual emotional availability of both the parent and child and are essential for child development. Whether parents’ emotional availability is influenced by their bonding history with their own parents and their current parenting stress has not been previously explored. The present study involved 29 father-child dyads and 36 mother-child dyads and evaluated their emotional availability during 10-min play session as well as current parenting stress and parents’ past care and overprotection experience. The manuscript addresses two important questions and uses appropriate statistical analyses to test the hypotheses. Findings from the pooled father-child and mother-child samples across all models incorporating bonding history and parenting stress indicated that fathers exhibited greater non-intrusiveness compared to mothers. In addition, regression slope analyses of the pooled sample showed that when difficult child score is low, higher maternal overprotection would predict greater adult non-hostility. In the mother-child sample, slope analyses showed that when difficult child score or parenting stress is low, higher paternal overprotection would predict greater adult sensitivity. The authors dealt with the discrepancy between these results and those of the existing literature by attributing it to cultural differences and also suggested alternative mechanisms for these interactions. Overall, the manuscript is written well and would benefit the field of parenting research.

Abstract:

  1. Lines 15-16: “Regression slope analyses showed that when parenting stress is low, higher parental overprotection would predict greater adult non-hostility”. Please be more accurate here, if this refers to the pooled sample “parenting stress” should be replaced with “difficult child score”, and “parental overprotection” with “maternal overprotection” as describe in the results.

>> Thank you for pointing this error out to us. We have amended the results section of the abstract accordingly.

“Healthy dyadic interactions serve as a foundation for child development and are typically characterised by mutual emotional availability of both the parent and child. However, several parental factors might undermine optimal parent-child interactions, including the parent’s current parenting stress levels and the parent’s past bonding experiences with his/her own parents. To date, no study has investigated the possible interaction of parenting stress and parental bonding history with their own parents on the quality of emotional availability during play interactions. In this study, 29 father-child dyads (18 boys, 11 girls; father’s age = 38.07 years, child’s age = 42.21 months) and 36 mother-child dyads (21 boys, 15 girls; mother’s age = 34.75 years, child’s age = 41.72 months) from different families were recruited to participate in a 10-min play session after reporting on their current parenting stress and past care and overprotection experience with their parents. We measured the emotional availability of mother-child and father-child play across four adult subscales (i.e. sensitivity, structuring, non-intrusiveness, non-hostility) and two child subscales (i.e. involvement and responsiveness). Regression slope analyses showed that parenting stress stemming from having a difficult child predicts adult non-hostility, and is moderated by the parents' previously experienced maternal overprotection. When parenting stress is low, higher maternal overprotection experienced by the parent in the past would predict greater non-hostility during play. This finding suggests that parents' present stress levels and past bonding experiences with their parents interact to influence the quality of the dyadic interaction with their child.”

Introduction:

  1. Line 90-91: “Despite such extensive literature….”. Please rewrite the sentence to clarify what “such extensive literature” refers to.

>> We have re-written this sentence and have provided some context to the literature we were referring to.

“Despite the considerable literature on parentalgbonding history and current parenting stress (e.g. [38,39,41,42,46,47]), studies examining their simultaneous effects in both father-child and mother-child dyads is lacking.”

  1. I would add a paragraph highlighting the importance of identifying bonding history and parenting stress as predictors for emotional availability in the sense that when a parent experiences difficulties (i.e., poor bonding history and/or high parenting stress) these can be targeted by psychological therapies and thereby can improve emotional availability when interacting with the child.

>> Thank you for this important feedback. We have illustrated the importance of identifying bonding history and parenting stress as predictors for emotional availability in the clinical context by adding the following in the Introduction section:

“From  the studies  above,  it  can  be seen  that  both parentalg bonding history  and  currently experienced  parenting stress  have  an impact  on  the emotional  availability  of the  parent  when interacting with their child.   Depending on the quality of parentalgbonding and parental stress, these changing parent-related variables will produce different levels of emotional availability during parent-child interaction. Identification of how specific patterns of parentalgbonding and parenting stress  affect  emotional availability  will  be clinically  significant  in improving  the  quality of  the parent-child relationship [48],  and provide a more nuanced approach to parenting interventions that have previously viewed parents as a relatively homogeneous group based on their life stages [49]. Psychotherapies that focus on improving parent-child interactions can take into account the prevailing relationships between parentalg bonding, parenting stress and emotional availability, and make use of specific strategies that target these factors to produce more efficacious outcomes [50].”

Discussion:

  1. Another paragraph should be added to suggest an explanation why overprotection was beneficial only in low score of difficult child and parenting stress. In other words, why is it different in higher scores where overprotection does not play a role in the association between stress and adult non-hostility/sensitivity.

>> Thank you for the feedback, we have added an explanation for why maternal overprotection, when interacting with higher scores of difficult child, has lower non-hostility scores.

“Conversely, results also indicate that lower adult non-hostility scores were observed when greater maternal overprotection scores interact with greater difficult child scores. This suggests that the advantage of parental non-hostility diminishes when parents perceive both greater maternal overprotection and greater stress in child management (as denoted by higher difficult child scores). An indication of high difficult child scores indicates a lower perceived ability to cope with parenting stress that stems from child management as well as they would like [59]. Although maternal overprotection may have been protective in parents with low difficult child scores (i.e., greater non-hostility observed in parenting behaviours), it is possible that maternal overprotection loses that effectiveness in parents who find their child difficult to manage in the first place. For example, children with difficult temperament has been shown to covary with greater maternal intrusiveness [78], lower maternal sensitivity [79] as well as greater parenting stress [80]. When parental coping resources are overwhelmed in this way, parents would then rely on prior experiences (i.e., parenting practices received when younger). Indeed, studies on intergenerational transmission of parenting showed that parents who perceived greater maternal control when younger are more likely to repeat similar power-assertive discipline methods when they become parents, especially so if they perceived their child as difficult to manage [37,81]. Through this, parenting stress associated with child management would be positively associated with maternal overprotection parents received when younger and display lower non-hostility scores as compared to parents with lower difficult child stress.”

  1. I would add a few sentences about the potential of psychological therapies in the cultural context in helping parents to resolve their issues around parental overprotection and thereby improve non-hostility/sensitivity when playing with their child. Likewise, parents who experience high parenting stress or perceive their child as difficult, regardless their bonding history, may also benefit from therapies aiming to reduce their stress by increasing their emotional availability.

>> Thank you for this comment. We have highlighted the important clinical implications of our study in the conclusion section:

“Finally, taking the results and the above implications discussed above, the findings of this study may be able to shed light on clinical applications on parenting skills and parent-child relationships in a culturally-specific context. It must be noted that parents from cultures where parental overprotection (or specifically in this study, maternal overprotection) is perceived more positively [67] may not benefit from parenting strategies that focus on cultivating child autonomy and independence at the expense of decreasing parental protection, as it disrupts the larger cultural influences of encouraging greater parental protection and involvement [75]. Instead, a more nuanced approach in teaching parenting skills that have to do with setting developmentally appropriate limits without undermining the child's potential to develop autonomy and independence [76] would be recommended.”

The manuscript would benefit from a close edit to correct typos and formatting and language errors (including symbols, numbers, punctuation, etc). For example, throughout the text many words are followed by the letter “g”.

>> We have checked the manuscript for typos, formatting and language errors, and have clarified what parentalgin our manuscript denotes:

“The prevailing literature supports the view that parentalgbonding history (parentalgdenotes the  intergenerational  influence of  past  parental bonding  history  on parents’  current  caregiving behaviours), which is evaluated from the parental care and overprotection that the parent received as a child, influence parent-child interactions in the subsequent generation [27–29].”